# Ornithine Aspartate and Vitamin-E Combination Has Beneficial Effects on Cardiovascular Risk Factors in an Animal Model of Nonalcoholic Fatty Liver Disease in Rats

**DOI:** 10.3390/biom12121773

**Published:** 2022-11-28

**Authors:** Laura Bainy Rodrigues de Freitas, Larisse Longo, Eduardo Filippi-Chiela, Valessa Emanoele Gabriel de Souza, Luiza Behrens, Matheus Henrique Mariano Pereira, Luiza Cecília Leonhard, Giulianna Zanettini, Carlos Eduardo Pinzon, Eduardo Luchese, Guilherme Jorge Semmelmann Pereira Lima, Carlos Thadeu Cerski, Carolina Uribe-Cruz, Mário Reis Álvares-da-Silva

**Affiliations:** 1Graduate Program in Gastroenterology and Hepatology, Universidade Federal do Rio Grande do Sul, Porto Alegre 90035-003, RS, Brazil; 2Experimental Laboratory of Hepatology and Gastroenterology, Center for Experimental Research, Hospital de Clínicas de Porto Alegre, Porto Alegre 90035-903, RS, Brazil; 3Center of Biotechnology, Universidade Federal do Rio Grande do Sul, Porto Alegre 91501-970, RS, Brazil; 4Department of Morphological Sciences, Universidade Federal do Rio Grande do Sul, Porto Alegre 90010-150, RS, Brazil; 5Unit of Surgical Pathology, Hospital de Clínicas de Porto Alegre, Porto Alegre 90035-903, RS, Brazil; 6Medical Genomics and Biotechnology Group, Universidad Católica de las Misiones, Posadas 3300, Argentina; 7Division of Gastroenterology, Hospital de Clínicas de Porto Alegre, Porto Alegre 90035-903, RS, Brazil

**Keywords:** animal model, cardiovascular risk, nonalcoholic fatty liver disease, ornithine aspartate, steatohepatitis, vitamin E

## Abstract

Cardiovascular (CV) disease is the main cause of death in nonalcoholic fatty liver disease (NAFLD), a clinical condition without any approved pharmacological therapy. Thus, we investigated the effects of ornithine aspartate (LOLA) and/or Vitamin E (VitE) on CV parameters in a steatohepatitis experimental model. Adult Sprague Dawley rats were randomly assigned (10 animals each) and treated from 16 to 28 weeks with gavage as follows: controls (standard diet plus distilled water (DW)), NAFLD (high-fat choline-deficient diet (HFCD) plus DW), NAFLD+LOLA (HFCD plus LOLA (200 mg/kg/day)), NAFLD+VitE (HFCD plus VitE (150 mg twice a week)) or NAFLD+LOLA+VitE in the same doses. Atherogenic ratios were higher in NAFLD when compared with NAFLD+LOLA+VitE and controls (*p* < 0.05). Serum concentration of IL-1β, IL-6, TNF-α, MCP-1, e-selectin, ICAM-1, and PAI-1 were not different in intervention groups and controls (*p* > 0.05). NAFLD+LOLA decreased miR-122, miR-33a, and miR-186 (*p* < 0.05, for all) in relation to NAFLD. NAFLD+LOLA+VitE decreased miR-122, miR-33a and miR-186, and increased miR-126 (*p* < 0.05, for all) in comparison to NAFLD and NAFLD+VitE. NAFLD+LOLA and NAFLD+LOLA+VitE prevented liver collagen deposition (*p* = 0.006) in comparison to NAFLD. Normal cardiac fibers (size and shape) were lower in NAFLD in relation to the others; and the inverse was reported for the percentage of regular hypertrophic cardiomyocytes. NAFLD+LOLA+VitE promoted a significant improvement in atherogenic dyslipidemia, liver fibrosis, and paracrine signaling of lipid metabolism and endothelial dysfunction. This association should be further explored in the treatment of NAFLD-associated CV risk factors.

## 1. Introduction

Metabolic-associated fatty liver disease (MAFLD), formerly called nonalcoholic fatty liver disease (NAFLD), comprises a spectrum of histological abnormalities, ranging from isolated steatosis to steatohepatitis, characterized by inflammation, necrosis, and hepatocellular ballooning, and progression to fibrosis, cirrhosis, liver failure and/or hepatocellular carcinoma [1,2,3]. It is the most common chronic liver disease worldwide, and massively affects health and economic systems [2,4,5]. The most frequent cause of death in patients with NAFLD is cardiovascular disease (CVD), accounting for about 40% of total deaths [6,7]. The overall number of patients with end-stage liver disease caused by NAFLD is rapidly increasing, and in the next decade, could become the leading cause of liver transplantation in the United States [8].

The underlying mechanisms leading to the close association between CVD and NAFLD are very complex and involve the activation of several different pathways, including abdominal obesity, atherogenic dyslipidemia, hypertension, dysglycemia, endothelial dysfunction, activation of oxidative stress, and inflammation, all together creating a pro-atherogenic environment favoring CVD development [9,10]. During this process, epigenetic changes also occur, mediated, for example, by microRNAs that act in the expression or suppression of genes responsible for the worsening of liver damage and CVD [11,12,13]. MicroRNAs are considered non-invasive biomarkers for early diagnosis of the lesion, as well as promising therapeutic targets [11,13]. This attention is undoubtedly warranted as it has important clinical implications for screening, surveillance, and treatment strategies of NAFLD-associated risk factors [9,10,14].

The cornerstone of NAFLD management remains lifestyle modification [9,10]. Weight loss, increased physical activity, and decreasing cardiometabolic risk factors have beneficial effects in NAFLD [9,10]. So far, there is no approved pharmacological treatment, however, numerous drugs are being evaluated in clinical protocols, either alone or in combination. Results are expected to be available within the next decade. Recently, it has been suggested that LOLA could be useful in controlling the NAFLD [15]. In this study, using an experimental steatohepatitis nutritional model that mimics the metabolic changes found in humans, we evaluated the risk of developing CVD and therapeutic effects of LOLA, alone or combined with vitamin E (VitE), a lipid-soluble chain-breaking antioxidant recommended in several guidelines for the treatment of NAFLD [14,16,17,18]. To this, we assessed the systemic inflammation, endothelial dysfunction, paracrine cell signaling the cardiomyocyte morphometry to explore the pathogenetic mechanisms underlying the association between NAFLD and CVD.

## 2. Materials and Methods

### 2.1. Animals and Study Design

Fifty adults (60-day old) male Sprague Dawley rats weighing 320–370 g were used. The animals were divided into groups of three or four per polypropylene cage with sawdust-covered floors, habituated in a controlled temperature environment (22 ± 2 °C) and a 12 h light/dark cycle. All experimental procedures were approved by the Ethics Committee for the Use of Animals (protocol #2019-0297) in accordance with international guidelines for animal welfare and measures were taken to minimize animal pain and discomfort.

After acclimatization, the animals were randomized into two experimental groups according to their weight. The control group (*n* = 10) received a standard diet (Nuvilab CR-1, Quimtia S.A.Paraná, Brazil) while the intervention groups (*n* = 40) received a high-fat and choline-deficient (HFCD) diet. The intervention groups received the following denomination: NAFLD (*n* = 10); NAFLD+LOLA (*n* = 10); NAFLD+VitE (*n* = 10) and NAFLD+LOLA+VitE (*n* = 10), as shown in Figure 1.

The diet of the intervention group was chosen because it recalls many of the phenotypes observed in humans with NAFLD, as demonstrated by our research group [14], being composed of 31.5% total fat, enriched with 54.0% trans fatty acids. The diet was replaced every two days. Both groups had water and food supplied *ad libitum* during the experiment period.

All animals were euthanized after 28 weeks of experiment and, prior to euthanasia, they were fasted for eight hours. The rats were anesthetized with isoflurane (Instituto BioChimico, Indústria Farmacêutica Ltd., Rio de Janeiro, Brazil) and euthanized by cardiac exsanguination. Blood samples were collected, centrifuged, and the serum obtained was maintained at −80 °C until the analyses were performed. After that, fragments of hepatic and cardiac tissue were fixed in 10% formaldehyde for the histopathological evaluation.

### 2.2. Ornithine Aspartate and Vitamin E Treatment

After sixteen weeks of standard diet or HFCD, therapeutic intervention was started daily with LOLA and/or VitE or the administration of vehicle (Veh) solution in the respective experimental groups, until euthanasia. We emphasize that the respective diet offered to the experimental groups remained available until the twenty-eighth week of the experiment. The administration of gavage started in the sixteenth week, as the animals in the intervention group had established liver damage. The animals in the control group and NAFLD received gavage daily with a Veh solution (0.5 mL/kg distilled water). The NAFLD+LOLA group received a daily dose of 200 mg/kg (LOLA; Biolab Sanus Farmacêutica Ltd., São Paulo, Brazil) [19,20]. The NAFLD+VitE group received treatment twice a week at a dose of 150 mg (VitE; Biolab Sanus Farmacêutica Ltd., Brazil), on the other days, it received gavage with Veh [17,21]. The NAFLD+LOLA+VitE group received the combination of the two drugs previously described in the same dose and frequency during the period.

### 2.3. Atherogenic Ratios

Serum levels of total cholesterol (TC), low-density lipoprotein-cholesterol (LDLc), high-density lipoprotein-cholesterol (HDLc), and triglycerides were determined using Labmax 560. The atherogenic ratios, calculated from the results of the lipid profile have been used as a tool for the prediction of cardiovascular risk. The atherogenic ratios were calculated as follows: Castelli’s Risk Index (CRI)-I = TC/HDLc, CRI-II = LDLc/HDLc and atherogenic coefficient (AC) = (TC-HDLc)/HDLc [22]. In this same equipment, alanine aminotransferase (ALT) and aspartate aminotransferase (AST) tests were performed to evaluate the liver function.

### 2.4. Systemic Inflammation and Endothelial Dysfunction

To detect serum changes in inflammation and endothelial dysfunction markers, we analyzed interleukin (IL)-1β, IL-6, tumor necrosis factor (TNF)-α, monocyte chemoattractant protein (MCP)-1, e-selectin, intercellular adhesion molecule (ICAM)-1, plasminogen activator inhibitor (PAI)-1, insulin, leptin, and adiponectina, using the multiplex assay of the Luminex platform (Millipore, Darmstadt, Germany). Serum evaluation of IL-10 was performed using the ELISA kit (Thermo Scientific, Waltham, MA, USA and MyBioSource, San Diego, CA, USA, respectively). The absorbance was measured in a spectrophotometer at a wavelength of 450 nm (Zenyth 200 rt). The results were expressed in ng/mL or pg/mL. All procedures were accomplished according to the manufacturer’s instructions, and all analyses were performed in duplicate.

### 2.5. Analysis of the Circulating microRNAs

To analyze the circulating microRNAs from serum, total RNA was extracted using the miRNeasy serum/plasma kit (Qiagen, Germantown, MD, USA). Then, cel-miR-39 (1.6 × 10^8^ copies) spike in control (Qiagen, USA) was added to provide an internal reference. cDNA conversion was performed from 10 ng of total RNA using the TaqMan microRNA Reverse Transcription kit (Applied Biosystems, Waltham, MA, USA). Analysis of the gene expression of miR-122, miR-33a, miR-126, miR-499, miR-186, and miR-146a together with the cell-miR-39 normalizer, was performed by RT-qPCR using TaqMan assay (Applied Biosystems, 2 Preston Ct Bedford, MA, USA). The sequences and codes of the assessed microRNAs are described in Appendix A. Values were calculated by the formula 2^−(ΔΔCt)^.

### 2.6. Liver Histopathological Analysis

Formalin-fixed liver tissue samples were embedded in paraffin and subjected to hematoxylin & eosin (H&E) and picrosirius red. Histopathological lesions of the different evolutionary stages of NAFLD were performed according to the score by Liang et al., which is a highly reproducible scoring system and applicable to the experimental models in rodents [23]. An experienced pathologist, blinded to the experimental groups, performed the analysis. Fibrosis was quantified by morphometric analysis after picrosirius red. Ten randomly selected field images were obtained per animal, using the Olympus BX51 microscope, and the QCapture X64 program with 200X magnification was used to determine staining intensity. This evaluation was performed using the ImageJ program (version 1.51p).

### 2.7. Cardiomyocytes Morphometric Analysis

Cardiomyocytes morphometric analysis (CMA) was performed based on adaptations of the nuclear morphometric analysis in tissue and this methodology has already been used by our research group [11,24,25]. Cardiomyocytes size and shape were measured using Image Pro Plus 6.0 Software (IPP6, Media Cybernetics, Rockville, MD, USA). H&E images from hearts of animals were acquired. Ten different fields were photographed to each animal using the QCapture X64 software in an Olympus BX51 microscope. At least 50 cross-sectioned cardiomyocytes of each animal were analyzed. The outline of single cells was marked using the magic wand tool from IPP6, followed by the acquisition of the following measurements: area, aspect, area/box, radius ratio, and roundness. These last four measurements were used to define the cardiomyocytes irregularity index (CII) to each cell (CII = aspect − area/box + roundness + radius ratio). Through these variables, we reported the size and shape of single cardiomyocytes. In addition to the averaged size and regularity, the plot of area versus CMA also defines the percentage of normal, hypertrophic, and atrophic cells. We also measured the variability of cardiomyocytes’ size and shape.

### 2.8. Statistical Analysis

Sample size estimation was performed using the WINPEPI 11.20 program (Brixton Health, Israel), based on a published study by the research group that demonstrated increased cardiovascular risk in an experimental model of NAFLD [14]. Data symmetry was tested using the Shapiro–Wilk test. Parametric data were performed by Analysis of Variance (ANOVA) followed by Tukey’s post hoc test and nonparametric data were analyzed using the Kruskal–Wallis followed by Dunn’s test. Quantitative variables were expressed as means ± standard deviations or medians and interquartile ranges (25th–75th). *p* < 0.05 was considered statistically significant. Data were analyzed in the Statistical Package for Social Sciences 28.0 (SPSS Inc., Chicago, IL, USA).

## 3. Results

### 3.1. General Characteristics

The general characteristics of the experimental model developed are shown in the Appendix A. The baseline body weights of the animals in all the experimental groups were similar (*p* = 0.999), demonstrating homogeneity. From the first week after the introduction of the HFCD, the four intervention groups showed a gradual increase in body weight, and a significant difference in these groups in comparison to the control group became evident after the second week of the experiment (*p* < 0.001). The animals in the NAFLD, NAFLD+LOLA, NAFLD+VitE, and NAFLD+LOLA+VitE experimental groups showed a significant increase in the body weight and fresh liver weight (*p* < 0.001) compared with those in the control group. There was no significant difference between the experimental groups for heart weight (*p* = 0.130). In the evaluation of liver function parameters, animals in the NAFLD group showed significantly higher serum levels of ALT (*p* < 0.001), compared with those in the control group. Serum AST levels were significantly lower in animals that received therapeutic intervention than those in the NAFLD group (*p* < 0.001).

### 3.2. Atherogenic Ratios to Assess Cardiovascular Risk

Regarding atherogenic ratios, the results obtained are shown in Figure 2A–C.

Animals in the NAFLD group showed a significant increase in the AC, CRI-I and CR-II (*p* < 0.001, for all) compared with the control group. There was no significant difference between the NAFLD group and the animals that received treatment alone with VitE or LOLA for the AC (*p* = 0.139 and *p* = 0.161, respectively), CRI-I (*p* = 0.139 and *p* = 0.161, respectively) and CRI-II (*p* = 0.246 and *p* = 0.200, respectively). However, animals treated with LOLA+VitE showed a significant decrease for AC, CRI-I and CRI-II (*p* < 0.001, for all) compared with the NAFLD group, leading to atherogenic rations that were like the control group (*p* = 1.00, *p* = 1.00 and *p* = 0.895, respectively). The results suggest that treatment with LOLA+VitE exerted a beneficial effect on atherogenic ratios.

### 3.3. Inflammation and Endothelial Dysfunction in the Assessment of Liver Damage and Cardiovascular Risk

Data obtained for the markers of inflammation and endothelial dysfunction are shown in Table 1. Regarding markers of endothelial dysfunction, we report a significant increase in MCP-1 concentration in the NAFLD+VitE group compared with the control group (*p* = 0.019). The concentration of e-selectin was significantly higher in animals treated with LOLA (*p* = 0.017) and LOLA+VitE (*p* = 0.035) compared with the control group. The groups treated with LOLA (*p* = 0.006), VitE (*p* = 0.042), and LOLA+VitE (*p* < 0.001) showed a significant increase in the concentration of ICAM-1 compared with the control group. There was a significant increase in insulin levels in animals from the NAFLD (*p* = 0.009) and NAFLD+LOLA+VitE (*p* < 0.001) compared with the control group. There was a significant increase in adiponectin concentration in the NAFLD (*p* = 0.006), NAFLD+LOLA (*p* = 0.003), and NAFLD+VitE (*p* = 0.001) groups compared with the control group. No significant differences were observed between the experimental groups for IL-1β, IL-6, TNF-α, IL-10, and PAI-1 (*p* > 0.05, for all).

### 3.4. Gene Expression of the Circulating microRNAs

The results obtained from the gene expression of the circulating microRNAs related to liver damage and cardiovascular risk are demonstrated in Figure 3A–F.

There was a significant increase in the gene expression of miR-122 in animals in the NAFLD group compared with the control group (*p* < 0.001). NAFLD+LOLA+VitE and NAFLD+LOLA groups showed a significant reduction in the expression of this marker in relation to the NAFLD (*p* < 0.001, in both) and NAFLD+VitE (*p* < 0.001, in both) groups (Figure 3A). miR-33a is significantly higher in the NAFLD group compared with the control group (*p* < 0.001). Among the groups of animals that received treatment, we did not report a significant difference between the NAFLD and NAFLD+VitE groups in the expression of circulating miR-33a (*p* = 0.506); however, there was a significant decrease in its expression when comparing the NAFLD and NAFLD+LOLA groups (*p* < 0.001) and the NAFLD and NAFLD+LOLA+VitE groups (*p* < 0.001), this expression being similar to the control group (*p* = 0.976, Figure 3B). Considering the expression of miR-126, the control group showed a significant increase in its expression compared with the NAFLD group (*p* = 0.004). The control group showed a gene expression similar to the NAFLD+LOLA+VitE group (*p* = 0.878); however, we observed a significant decrease in miR-126 expression in the NAFLD+LOLA (*p* < 0.001) and NAFLD+VitE (*p* < 0.001) groups, this expression being similar to the NAFLD group (*p* = 0.856 and *p* = 0.833, respectively, Figure 3C). The control group had miR-499 expression similar to the NAFLD and NAFLD+LOLA groups (*p* = 0.900 and *p* = 0.730, respectively). The expression of this gene was significantly higher in the NAFLD+VitE and NAFLD+LOLA+VitE groups compared with the control and NAFLD groups (*p* = 0.009 and *p* = 0.014, respectively, Figure 3D). The NAFLD group showed a significant increase in miR-186 expression compared with the control group (*p* = 0.003). The groups that received treatment with LOLA and LOLA+VitE had similar gene expression of mir-186 in relation to the control group (*p* = 0.925 and *p* = 1.00, respectively), however, the animals treated with VitE presented a significant increase in the expression of this marker in relation to the control group (*p* = 0.043, Figure 3E). There was no difference between groups in the expression of miR-146a (*p* > 0.05, Figure 3F).

### 3.5. Morphometric and Histopathological Evaluation of Cardiomyocytes

Myocardial steatosis was not observed in any of the experimental groups. The characterization of cardiomyocytes’ morphometry (i.e., size and shape) provides valuable information about the myocardium function [26]. Images obtained from the slides stained with H&E are presented in Appendix A. To assess this, we measured the size and shape of cardiomyocytes (Figure 4A–E).

Among the groups, NAFLD presented the lowest percentage (67%) of fibers with normal morphometry, and the groups that received treatment with LOLA (76%), VitE (81%), and LOLA+VitE (75%) had a similar percentage of normal fibers to the control group (73%). However, no significant difference was observed between the experimental groups for this variable (*p* = 0.114). The percentage of regular hypertrophic cardiomyocytes was higher in the NAFLD group (26%) compared with the control groups (16%), NAFLD+LOLA (16%), NAFLD+VitE (14%), and NAFLD+LOLA+VitE (17%), but no significant differences were observed between the groups (*p* = 0.229). The percentage of regular hypertrophic fibers were lower in the NAFLD+LOLA (2%) and NAFLD+VitE (1%) groups compared with the other experimental groups, with no significant difference (*p* = 0.075). The percentage of irregular atrophic, regular atrophic, and irregular hypertrophic fibers was similar between the experimental groups (*p* > 0.05). No significant difference (*p* = 0.569) was observed between the experimental groups for the CII (Figure 5A). The variability of cardiomyocytes’ area (Figure 5B) was significantly lower in the VitE group compared with the other experimental groups (*p* = 0.040). Altogether, these results suggest that there are no significant changes in the histological structure of the myocardium in the HFCD diet-induced steatohepatitis model.

### 3.6. Liver Histopathological Analysis

Images obtained from the slides stained with H&E and picrosirius are presented in Appendix A. No abnormalities were seen in the liver tissue of the control group (Appendix A), whereas the animals in the NAFLD, NAFLD+LOLA, NAFLD+VitE and NAFLD+LOLA+VitE groups had predominantly microvesicular steatosis along with macrovesicular steatosis of moderate intensity, with inflammatory activity and mild hypertrophy (Appendix A). In the staging of the histopathological lesion, six animals in the NAFLD and NAFLD+LOLA groups developed simple steatosis and four animals presented steatohepatitis. In the experimental groups NAFLD+VitE and NAFLD+LOLA+VitE, seven animals presented simple steatosis and three animals developed steatohepatitis. No hepatic histopathological changes were observed for the control group. In the quantification of collagen, using picrosirius-staining showed a significantly greater amount of connective tissue fibers in the NAFLD group (0.99 ± 0.44) compared with the control (0.45 ± 0.23; *p* = 0.009), NAFLD+LOLA (0.37 ± 0.20; *p* = 0.002), and NAFLD+LOLA+VitE (0.42 ± 0.27; *p* = 0.006) groups. There was no significant difference (*p* = 0.764) in collagen fiber deposition between the NAFLD and NAFLD+VitE groups (0.81 ± 0.50).

## 4. Discussion

Although NAFLD has recently been renamed MAFLD, there are important differences between the two clinical entities [2,27]. As the largest body of evidence associating liver steatosis with cardiovascular risk comes from the NAFLD population, this was the nomenclature chosen for this study. NAFLD and CVD are both associated with metabolic risk factors; however, the underlying mechanisms linking both clinical conditions are complex and simultaneously involve several pathways. This study provided additional evidence linking NAFLD to the development of CVD, since animals in the NAFLD group presented at the same time more significant liver disease, including a higher liver collagen deposition, and features of metabolic disease, with marked adipokines and insulin abnormalities, higher atherogenic ratios and paracrine disbalance, as well as a lower number of normal cardiomyocytes in comparison to controls. Additionally, animals treated with LOLA+VitE showed a significant reduction in atherogenic indexes (AC, CRI-I, and CR-II) compared to the NAFLD-group. Treatment with LOLA alone or associated with VitE, demonstrated benefit on paracrine signaling, significantly decreasing the gene expression of miR-122, miR-33a and miR-186 compared with NAFLD animals. Additionally, treatment with LOLA+VitE promoted a significant increase in miR-126 expression, demonstrating a positive synergistic effect on signaling pathways related to liver injury and endothelial dysfunction.

The progression of NAFLD results from an imbalance between lipid uptake and lipid disposal and eventually causes oxidative stress and hepatocyte injury [10]. These factors together create a pro-atherogenic environment favoring CVD development [10]. In this study, we demonstrated that abnormalities of lipid metabolism and atherogenic ratios were related to greater susceptibility to develop CVD associated with steatohepatitis. These data corroborate with previous results obtained by the research group, in which we reported significant correlations between the presence of atherogenic dyslipidemia, systemic inflammation, endothelial dysfunction, liver fibrogenesis, and gut dysbiosis, factors that contribute to NAFLD progression and increased cardiovascular risk [11,27]. Atherogenic dyslipidemia, often present at NAFLD, is characterized by plasma hypertriglyceridemia, increased small dense LDLc particles, and decreased HDLc levels [28]. Here, LOLA+VitE was effective in preventing atherogenic dyslipidemia. Limited evidences from clinical trials support the thesis that LOLA has hepatoprotective properties in patients with NAFLD/nonalcoholic steatohepatitis. Such evidence includes the ability of LOLA to attenuate raised levels of liver enzymes including alanine aminotransferase and to reduce serum triglycerides [29,30,31]. In the PIVENS study population, vitamin E treatment demonstrated that resolution of steatohepatitis is associated with improvements in triglycerides and HDL, but not in other CVD risk factors, including LDL levels [18,32]. To our knowledge, there are no studies evaluating the association of LOLA+VitE treatment and its relationship to the cardiovascular risk in steatohepatitis. However, based on the above, we can infer that the treatment with LOLA+VitE was beneficial for promoting the improvement of atherogenic dyslipidemia, a finding constantly observed in NAFLD cases. In this sense, it is important that the new treatments for steatohepatitis carry out the evaluation of the potential beneficial and deleterious effects also related to prevention in the development of CVDs.

In this study, we did not report significant differences between the NAFLD group and the groups that received treatment for markers of systemic inflammation and endothelial dysfunction. However, we report a beneficial effect in the associated treatment, NAFLD+LOLA+VitE group under paracrine signaling mediated by miR-122, miR-33a, miR-186, and miR-126 in relation to the NAFLD group and NAFLD+VitE group. Along with genetic predisposition to NAFLD and CVD, disease development and natural history are also modulated by epigenetic mechanisms, including the expression of microRNAs, which are small particles of non-coding RNA with specific functionality [12,13]. Epigenetic modifications affect hepatic lipid metabolism, insulin resistance, mitochondrial function, oxidative stress, and therefore, the assessment of microRNAs has been used for early detection and monitoring of NAFLD progression, and to assess clinical and subclinical CVD [11,12,13]. The miR-122 is the most abundant in the liver and plays a fundamental role in liver physiology and lipid metabolism, together with mir-33a [12,33]. The miR-186 and miR-126 play a key role in the inflammatory process involving endothelial cell dysfunction [11,34]. The miR-499 is muscle-specific, expressed mainly by quiescent cardiomyocytes, and acts by targeting cell death, thus being a biomarker of the severity and extent of cardiac injury [35,36]. Decreased expression of miR-146a is associated with a higher risk of developing atherosclerosis, acting in the regulation of the inflammatory response [37,38]. The data obtained for these microRNAs in this study corroborate previous results described on NAFLD in relation to healthy controls and can be considered an extrahepatic fingerprint of steatohepatitis [12,14,33,39]. Additionally, we emphasize that although we did not observe a significant difference between the groups in the protein concentration of inflammatory markers and endothelial diffusion, we reported significant differences between the control group and NAFLD+LOLA+VitE group in relation to the NAFLD group for the expression of miR-186 and miR-126. Both are vital regulators of multiple proteins involved in this process and are associated with a global dysmetabolic disease state and CVD risk. Regarding VitE, it was shown that rats fed a VitE-deficient diet decreased the expression of miR-122 and miR-125 compared with rats fed a VitE-deficient diet, which results in the alteration of lipid metabolism and inflammation [30]. There are no studies in the literature that demonstrate the effect of LOLA treatment on the expression of microRNAs. We emphasize that in this study, a beneficial effect of the synergistic treatment with LOLA+VitE was observed on the expression of microRNAs related to the regulation of cholesterol, inflammation, lipid metabolism, and endothelial dysfunction, contributing to the protection against the development of metabolic disorders and CVD related to steatohepatitis.

Several cellular processes can be inferred through morphometric analysis and this methodology can be used in the diagnosis and prognosis of some clinical conditions [24,40,41]. An interesting analysis carried out in this study was the morphometric evaluation of cardiomyocytes, in which we reported that animals with NAFLD had lower percentages of cardiac fibers with normal morphometric appearance compared with the other experimental groups. The inverse was observed for the percentage of regular hypertrophic cardiomyocytes. Morphological characteristics of muscle fibers, such as fiber size, are critical factors that determine the health and function of the muscle. Morphometric alterations of muscle fibers may occur as a result of the imbalance between protein synthesis and proteolysis, due to metabolic alterations [42]. From a threshold of adaptability and intensity, these cellular changes may present the risk of altering the structure and functioning of the tissue. Hypertrophy occurs as a result of an increased rate of protein synthesis characterized by an increase in the size of individual myofibers, and may be triggered by pathological conditions such as cardiovascular disorders [43]. This rationale corroborates the data obtained in this study regarding the higher percentage of regular hypertrophic cardiomyocytes in the NAFLD-group compared with the others, since this group is more likely to develop CVD associated with steatohepatitis. Imaging methods that detect changes in heart structures, such as the electrocardiogram, have diagnostic potential. The absence of this assessment in the study is a limitation, as we would be able to adequately characterize and/or identify the presence or suspicion of CVD in animals with NAFLD.

Despite the growing public health impact of steatohepatitis, treatment options remain limited and there are no FDA-approved therapies [44,45]. One of the reasons for the absence of uncontroversial effective pharmacotherapy for steatohepatitis is the high complexity of this disease, caused by multiple interacting factors [29]. Accordingly, well-designed experimental models and clinical studies are useful to elucidate the pathophysiological mechanisms related to the development and progression of the disease, as well as to develop effective therapeutic strategies [46]. In the literature, a limited number of studies have evaluated the use of LOLA in patients with fatty liver of diverse etiology, highlighting the relevance of the present study [15,47]. The key mechanism of action of LOLA involves the removal of ammonia via two distinct mechanisms, namely the synthesis of urea (L-ornithine is a metabolic intermediate in the urea cycle) by periportal hepatocytes and the synthesis of glutamine via glutamine synthetase, an enzyme located in both perivenous hepatocytes and skeletal muscle [15,47]. Butterworth et al. pointed out that the use of LOLA is effective for NAFLD; however, additional studies should be carried out [15]. Vitamin E, in its different forms, is a pleiotropic agent with effects in virtually all cellular components and biological fluids of humans. In clinical practice, the administration of VitE (800 IU/day) was tested along with pioglitazone and placebo in non-diabetic and non-cirrhotic patients with steatohepatitis, with a histological improvement of the lesion demonstrated by resolution of steatohepatitis in 36% of the patients who received VitE treatment. However, there are considerable safety concerns related to this treatment, including general mortality and hemorrhagic stroke. Our results demonstrate the beneficial effect of treatment with LOLA, alone or in combination with VitE, on the deposition of collagen fibers in the liver. Many pharmacological treatments are being evaluated through clinical trials with the primary objective of improving liver fibrosis [29,48,49]. Strategies to combine drugs are largely empirical at present, with most combinations seeking to include metabolic targets with either antifibrotic and/or anti-inflammatory agents. There are no data published evaluating the isolated and synergistic effect of LOLA and/or VitE in relation to NAFLD and the risk of developing CVD, justifying further studies. The beneficial effect of the isolated use of LOLA or the association of LOLA+VitE in decreasing the collagen fiber deposition should be further investigated in preclinical and clinical protocols, to elucidate whether this would be a viable treatment for NAFLD.

NAFLD patients deserve a thoughtful cardiovascular risk assessment and evaluation for subclinical atherosclerosis, as changes in lipid and lipoprotein metabolism, along with the development of an inflammatory process and endothelial dysfunction, are the major contributing factors linking NAFLD to CVD [10,28]. Given the current lack of approved pharmacological therapies for steatohepatitis, studies are needed to better understand the possible mechanisms and, therefore, the therapeutic management that derives from them [10,28,44]. In conclusion, we demonstrated that combined treatment with LOLA+VitE in NAFLD prevented changes in atherogenic dyslipidemia, possibly mediated by paracrine regulation of microRNAs that act on lipid metabolism, inflammation, and endothelial dysfunction. Additionally, this treatment promoted less deposition of collagen fibers in the liver tissue. Given the above, the use of LOLA+VitE on cardiovascular risk parameters in NAFLD should be further explored to understand the underlying factors related to the process.

## Figures and Tables

**Figure 1 biomolecules-12-01773-f001:**
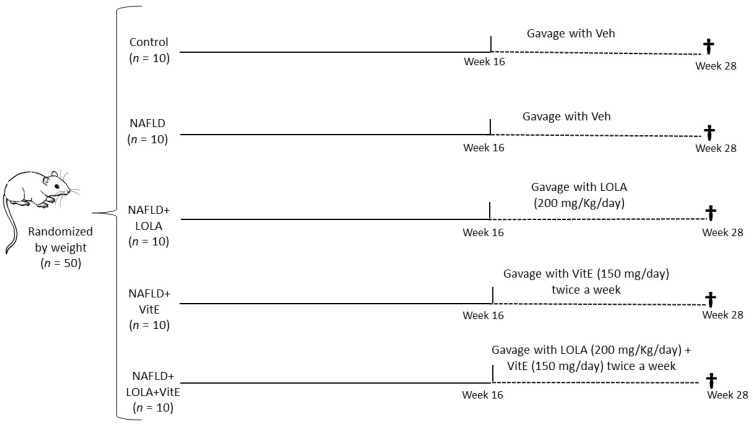
Experimental Design. The control group (*n* = 10) received a standard diet and the four intervention groups (*n* = 40) received a high-fat and choline-deficient (HFCD) during the 28-week experiment. After sixteen weeks, the control group and NAFLD received gavage daily with a Veh solution; NAFLD+LOLA group received a daily dose of 200 mg/kg; NAFLD+VitE group received twice a week at a dose of 150 mg; and NAFLD+LOLA+VitE group received the combination of the two drugs. After all, the animals were euthanized (†). Abbreviations: LOLA: ornithine aspartate; NAFLD: nonalcoholic fatty liver disease; Veh: vehicle, and VitE: vitamin E.

**Figure 2 biomolecules-12-01773-f002:**
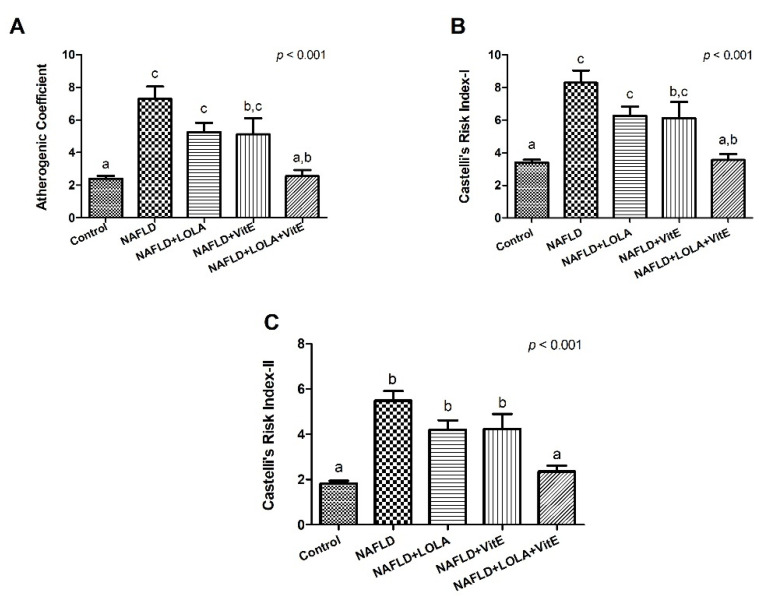
Atherogenic ratios. (**A**) Atherogenic coefficient (AC), (**B**) Castelli’s Risk Index (CRI)-I, and (**C**) CRI-II. Data expressed as mean ± standard deviation, Tukey’s test. Different letters indicate a significant difference between groups (*p* < 0.05). LOLA: ornithine aspartate; NAFLD: nonalcoholic fatty liver disease, and VitE: vitamin E.

**Figure 3 biomolecules-12-01773-f003:**
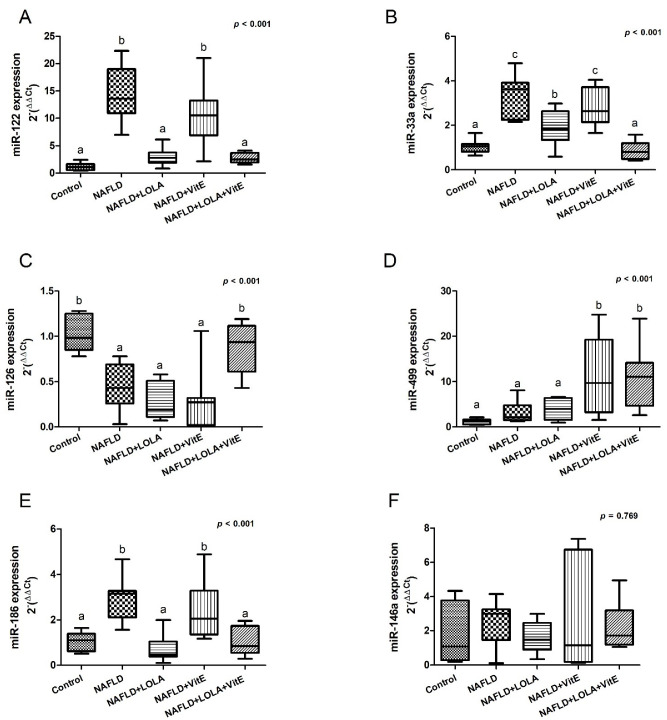
Gene expression of the circulating microRNAs. (**A**) miR-122, (**B**) miR-33a, (**C**) miR-126, (**D**) miR-499, (**E**) miR-186, and (**F**) miR-146a. Data expressed as median (25th–75th percentile), Kruskal–Wallis test. Different letters indicate a significant difference between groups (*p* < 0.05). LOLA: ornithine aspartate; NAFLD: nonalcoholic fatty liver disease, and VitE: vitamin E.

**Figure 4 biomolecules-12-01773-f004:**
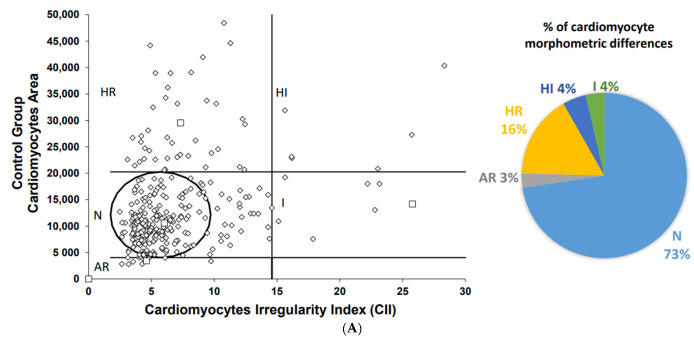
Cardiomyocytes morphometric analysis. The area and cross-sectional shape of cardiomyocytes were determined from images of hematoxylin and eosin-stained tissue. Dot plot of cardiomyocytes area vs. cardiomyocytes irregularity index in (**A**) Control; (**B**) NAFLD; (**C**) NAFLD+LOLA; (**D**) NAFLD+VitE, and (**E**) NAFLD+LOLA+VitE. Each dot represents a population of cardiomyocytes with different morphometry. Data expressed in percentage (%). AR: atrophic regular; HI: hypertrophic irregular; HR: hypertrophic regular; N: normal area and shape, I: irregular area and shape.

**Figure 5 biomolecules-12-01773-f005:**
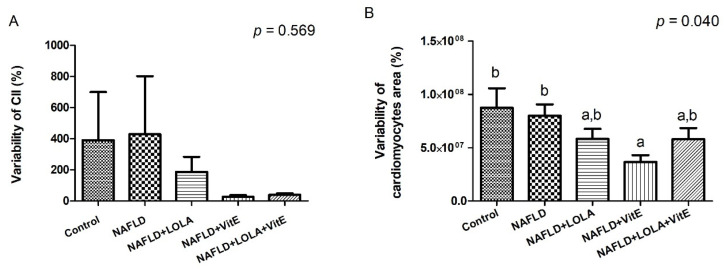
Cardiomyocytes morphometric analysis. (**A**) Cardiomyocytes irregularity index (CII); (**B**) variability of cardiomyocyte area. Data expressed as median (25th–75th percentile), Kruskal–Wallis test. Different letters indicate a significant difference between groups (*p* < 0.05). LOLA: ornithine aspartate; NAFLD: nonalcoholic fatty liver disease, and VitE: vitamin E.

**Table 1 biomolecules-12-01773-t001:** Inflammation and endothelial dysfunction in a nutritional model of nonalcoholic fatty liver disease.

Variables ^#^	Control	NAFLD	NAFLD+LOLA	NAFLD+VitE	NAFLD+LOLA+VitE	*p **
IL-1β (pg/mL)	7.92 (0.69–15.16)	5.50 (1.49–9.51)	7.87 (2.0−13.73)	9.04 (3.30–14.79)	10.12 (1.17–19.06)	0.843
IL-6 (pg/mL)	191.75 (36.93–994.25)	116.15 (13.96–439.46)	90.94 (25.47–149.70)	113.67 (13.96–217.46)	22.20 (1.96–994.25)	0.165
TNF-α (pg/mL)	5.21 (1.44–8.99)	3.49 (2.71–4.25)	4.02 (2.58−5.46)	4.56 (3.28–5.85)	3.39 (1.90–4.87)	0.580
IL-10 (pg/mL)	33.17 (19.93–49.62)	56.07 (29.12–91.3)	44.66 (24.88–76.66)	53.55 (29.12–76.84)	48.05 (27.53–66.41)	0.062
MCP-1 (pg/mL)	317.39 (252.11–382.66) ^a^	433.32 (356.65–510.0) ^a,b^	395.64 (259.58–531.70) ^a,b^	523.19 (450.96–595.42) ^b^	453.10 (317.08–588.92) ^a,b^	0.037
e-selectin (ng/mL)	1.72 (1.42–2.03) ^a^	2.38 (1.95–2.80) ^a,b^	2.52 (2.22–2.82) ^b^	2.20 (1.76–2.65) ^a,b^	2.43 (2.0–2.87) ^b^	0.015
ICAM-1 (ng/mL)	0.14 (±0.03) ^a^	0.46 (±0.15) ^a,b^	0.67 (±0.15) ^b^	0.60 (±0.18) ^b^	1.06 (±0.40) ^b^	<0.001
PAI-1 (pg/mL)	42.56 (18.24–66.88)	46.28 (29.04–63.52)	106.26 (29.57–182.94)	89.25 (43.46–135.03)	122.80 (19.78–225.80)	0.170
Insulin (pg/mL)	381.52 (±74.72) ^a^	1750.39 (±498.76) ^b^	1170.23 (±264.63) ^a,b^	1023.09 (±204.90) ^a,b^	1880.82 (±326.98) ^b^	<0.001
Leptin (pg/mL)	2227.51 (68.93–7502.01) ^a^	10119.70 (1203.67–25894.03) ^b^	7240.19 (3879.69–10600.70) ^a,b^	10807.54 (2779.80–18835.28) ^b^	11202.96 (7618.65–14787.270) ^a^	0.025
Adiponectina (ng/mL)	15.32 (11.36–19.28) ^a^	35.45 (26.87–44.03) ^b^	37.31 (29.59–45.04) ^b^	37.91 (22.91–52.92) ^b^	30.05 (26.10–35.25) ^a,b^	0.001

^#^ Variables expressed by mean (±standard deviation), or median (25th–75th percentiles). * *p* < 0.05 is considered significant, different letters indicate a significant difference between groups. Abbreviations: IL: interleukin, ICAM: intercellular adhesion molecule, LOLA: ornithine aspartate, MCP: monocyte chemoattractant protein, NAFLD: nonalcoholic fatty liver disease, PAI: plasminogen activator inhibitor, TNF: tumor necrosis factor, and VitE: vitamin E.

## Data Availability

Not applicable.

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
