# Peer review of "Ornithine Aspartate and Vitamin-E Combination Has Beneficial Effects on Cardiovascular Risk Factors in an Animal Model of Nonalcoholic Fatty Liver Disease in Rats"

_biomolecules, 2022, doi:10.3390/biom12121773_

Round 1
Reviewer 1 Report
1. Please list the evidence or citation of the feeding environment of rats.
2. What kind of statistical methods determine you to use the number of rats?
3. All of your results focused on morphometric and histopathology, did the physical function considered? For example, ultrasound is an appropriate method to evaluate the pump function and pathological changes non-invasively.
4. Why the microRNAs were introduced here?
5. Why did you choose Ornithine Aspartate combined with VitE to perform this research?
Author Response
- Please list the evidence or citation of the feeding environment of rats.
Answer: We appreciate your comment. For the development of this study, the animals of the four intervention groups (NAFLD, NAFLD+LOLA, NAFLD+VitE and NAFLD+LOLA+ VitE) received a high-fat choline-deficient diet (HFCD). This diet has already been used in previous studies published (doi: 10.2147/CEG.S262879 and DOI: 10.4254/wjh.v13.i12.2052) by our research group and is capable of inducing NAFLD in rats, like the disease phenotype observed in humans. During the study, the animals had free access to food. This information is included in the Animals and Study Design.
- What kind of statistical methods determine you to use the number of rats?
Answer: Information on sample size calculation was added to the topic "Statistical Analysis" in the new version of the manuscript.
- All of your results focused on morphometric and histopathology, did the physical function considered? For example, ultrasound is an appropriate method to evaluate the pump function and pathological changes non-invasively.
Answer: We appreciate your comment and agree with your inference. The assessment of the presence of possible arrhythmias in an electrocardiographic study, in addition to the pump function, would be a significant finding for this study, but this would entail the use of resources that were not available at the time in the experimental laboratory. A succinct comment was added in the Discussion Section – Study limitations.
- Why the microRNAs were introduced here?
Answer: We appreciate your question. These molecules, regulated by epigenetic processes, are responsible for promoting the expression and/or suppression of genes that can worsen liver damage and increase cardiovascular risk. In this study, we selected, based on the literature, some microRNAs involved in this process to be evaluated. Information on microRNAs was pending and was added to the manuscript's introduction and discussion.
- Why did you choose Ornithine Aspartate combined with VitE to perform this research?
Answer: Vitamin E is an established treatment for NAFLD and its use is authorized in cases of steatohepatitis in several international consensuses (ALEH, AASLD, EASL), but with limited results, which makes it necessary to look for other therapeutic alternatives for the disease. There are numerous drugs currently being evaluated in various international studies, either alone or in combination. Results are expected to be available by the end of this decade. Preliminary evidence suggests that ornithine aspartate (LOLA) may be useful in NAFLD, which motivated its testing in this experimental model. We decided to include a combination group to assess the synergistic effect of the two drugs, as they have different targets in the pathogenesis of NAFLD. New information has been added to the introduction and discussion topic of the manuscript.

Author Response
- The authors should introduce what ornithine aspartate is and its function within the context of existing literature. Also, why did the authors choose Vitamin E in the combination treatment for this study? In addition, why did the authors measure miRNAs? A brief introduction should be included in the manuscript.
Answer: We appreciate your comment and agree with your inference. New information regarding ornithine aspartate (LOLA) was included in the manuscript discussion and we emphasize the fact that there is little evidence of its use in NAFLD. Vitamin E is an established treatment for NAFLD and its use is authorized in cases of steatohepatitis in several international consensuses (ALEH, AASLD, EASL), but with limited results, which makes it necessary to look for other therapeutic alternatives for the disease. There are numerous drugs currently being evaluated in various international studies, either alone or in combination. Results are expected to be available by the end of this decade. Preliminary evidence suggests that ornithine aspartate may be useful in NAFLD, which motivated its testing in this experimental model. We decided to include a combination group to assess the synergistic effect of the two drugs, as they have different targets in the pathogenesis of NAFLD. Regarding microRNAs, we added new comments in the introduction and discussion of the manuscript, justifying the analysis.
- Figure 1. Because all four intervention groups received a high-fat and choline-deficient diet only for the initial 16 weeks, the authors should reorganize and label the figure to better illustrate the experimental design, such as using “NAFLD+LOLA” instead of “LOLA”.
Answer: We agree with the reviewer's suggestion. The names of the experimental groups were modified throughout the manuscript for standardization, that is, the groups that received treatment were now called NAFLD+LOLA, NAFLD+VitE and NAFLD+LOLA+VitE. Additionally, in the topic entitled “Ornithine Aspartate and Vitamin E Treatment” we clarified that the intervention groups (NAFLD, NAFLD+LOLA, NAFLD+VitE and NAFLD+LOLA+VitE) received a high-fat and choline-deficient diet during the twenty-eight weeks of the experiment, the treatment with the respective medication offered to the experimental group started in the sixteenth week of the experiment until euthanasia.
- Did the authors observed any changes in body, liver and heart weights of mice that were in different intervention groups? What about serum levels of ALT and AST?
Answer: Information on the general characteristics of the experimental model and the serum levels of AST and ALT were added in the new version of the manuscript (topic General Characteristics in the results).
- Figure3. The authors stated that LOLA+VitE treatment “significantly decreasing the gene expression of miR-122, miR-33a and miR-186 compared to NAFLD animals”. However, for miR-122 and miR-186, there were no differences between LOLA only group and the combination treatment group, suggesting the combination treatment did not provide further beneficial effect in the animal model. Why did the authors measure miR-499 and miR-146a?
Answer: The sentence inserted in the manuscript discussion has been reworded. Additionally, we insert the justification for evaluating the expression of miR-499 and miR-146a, both microRNAs act in the process of cell death and inflammatory response in cardiac injury.
- Figure 4. Images of cardiomyocytes’ morphology should be included. Data labels for the pie chart is needed.
Answer: We agree with the reviewers' suggestion. Histopathological images of cardiomyocyte analysis were added in supplementary data. We perform the addition of the pending information in the pie charts (figure 4).
- Hepatic H&E staining should be included for liver histopathological analysis. s-SMA staining should be included for hepatic fibrosis.
Answer: The images obtained from the histopathological analysis of liver tissue that were stained with H&E and picrosirius were added in Supplementary Data.
- In the discussion section, please clarify the sentence “These findings were more pronounced with the association than with the isolated drugs”.
Answer: We agree with the reviewer. Upon re-reading the manuscript, we noticed that this sentence is confusing and we excluded it from the first line of discussion in the manuscript.
- Please also read over very carefully to check for grammatical errors and consistency.
Answer: We appreciate your observation. We revised the manuscript in English and made adaptations

Round 2
Reviewer 1 Report
After the first round of revisions, I agree this version of manustript to be published.
Author Response
We appreciate the reviewer-1 suggestions that were offered in the first review round. These notes contributed to the improvement of the manuscript.
Reviewer 2 Report
First, the reviewer appreciates the authors for revising the manuscript accordingly.
Just a few more questions to address:
1. Regarding "General Characteristics", a data table is needed to show all the measurement results. Adding only description to the result section is not enough.
2. For supplementary figures, each image should be labeled with specific treatment. The size of all images should be consistent. The letter within each image should be the same size as well.
3. For supplementary figure 2 and 3, the same magnification should be applied to all images. Please indicate which supplementary figure is described within the results section.
4. In the results section 3.6, please move the last sentence of the paragraph to the beginning.
Author Response
REVIEWER 2
- Regarding "General Characteristics", a data table is needed to show all the measurement results. Adding only description to the result section is not enough.
Answer: We appreciate your comment and agree with your inference. The data description was added in supplementary table 2.
- For supplementary figures, each image should be labeled with specific treatment. The size of all images should be consistent. The letter within each image should be the same size as well.
Answer: The supplementary images were reformulated and labeled with the name of the experimental group, as suggested by the reviewer. The font and size of the images were standardized
- For supplementary figure 2 and 3, the same magnification should be applied to all images. Please indicate which supplementary figure is described within the results section.
Answer: The same magnification was applied to all supplementary images and information about which supplementary image is cited in the result was added to the manuscript.
- In the results section 3.6, please move the last sentence of the paragraph to the beginning.
Answer: As suggested by the reviewer, the last sentence of the paragraph of the results described in section 3.6 "Images obtained from slides stained with H&E and picrosirius is presented in Supplementary Data" was moved to the beginning of the description